# Associations of remote mental healthcare with clinical outcomes: a natural language processing enriched electronic health record data study protocol

Muhammad Shamim Ahmed [1], Daisy Kornblum,[2] Dominic Oliver [1,3,4] Paolo Fusar-Poli [1,5] Rashmi Patel [2,6]

MSA and RP contributed equally.

For numbered affiliations see end of article.

**Correspondence to**
Dr Rashmi Patel;
bmj@rpatel.co.uk

## ABSTRACT

**Introduction** People often experience significant difficulties in receiving mental healthcare due to insufficient resources, stigma and lack of access to care. Remote care technology has the potential to overcome these barriers by reducing travel time and increasing frequency of contact with patients. However, the safe delivery of remote mental healthcare requires evidence on which aspects of care are suitable for remote delivery and which are better served by in-person care. We aim to investigate clinical and demographic associations with remote mental healthcare in a large electronic health record (EHR) dataset and the degree to which remote care is associated with differences in clinical outcomes using natural language processing (NLP) derived EHR data.

**Methods and analysis** Deidentified EHR data, derived from the South London and Maudsley (SLaM) National Health Service Foundation Trust Biomedical Research Centre (BRC) Case Register, will be extracted using the Clinical Record Interactive Search tool for all patients receiving mental healthcare between 1 January 2019 and 31 March 2022. First, data on a retrospective, longitudinal cohort of around 80 000 patients will be analysed using descriptive statistics to investigate clinical and demographic associations with remote mental healthcare and multivariable Cox regression to compare clinical outcomes of remote versus in-person assessments. Second, NLP models that have been previously developed to extract mental health symptom data will be applied to around 5 million documents to analyse the variation in content of remote versus in-person assessments.

**Ethics and dissemination** The SLaM BRC Case Register and Clinical Record Interactive Search (CRIS) tool have received ethical approval as a deidentified dataset (including NLP-derived data from unstructured free text documents) for secondary mental health research from Oxfordshire REC C (Ref: 18/SC/0372). The study has received approval from the SLaM CRIS Oversight Committee. Study findings will be disseminated through peer-reviewed, open access journal articles and service user and carer advisory groups.

## STRENGTHS AND LIMITATIONS OF THIS STUDY

⇒ The use of electronic health record data will enable comparisons between remote versus in-person assessments in a large sample of National Health Service patients with mental health disorders which is representative of real-world clinical practice.

⇒ Natural language processing models will enable analysis of free text electronic health record data to ascertain differences in clinical presentation, assessment and outcomes between patients receiving remote and in-person assessments.

⇒ Routinely recorded data will be investigated for secondary research purposes. However, data on clinical outcomes may be missing or inaccurate. Data availability relies on the recording of accurate data at the point of clinical contact.

⇒ As this is an observational study, associations with remote mental healthcare could be confounded by indication and it will not be possible to infer any causality between observed associations with clinical outcomes.

⇒ The study will analyse secondary mental health data which does not include data from primary care.

## INTRODUCTION

Remote technologies such as phone, video calls, text messages and email allow patients and clinicians to communicate with each other when it is not possible to meet in-person. The COVID-19 pandemic accelerated the implementation of remote care technology in mental health with services often using remote technology to maintain clinical contact and deliver care.[1 2] Although restrictions resulting from the pandemic have eased, remote technology is likely to continue to be used and remain a useful adjunct to conventional in-person consultation.[3]

Access to conventional mental healthcare may be hindered due to lack of resources, stigma and lack of access to community mental health services.[1] Remote care technology has the potential to overcome these barriers and augment in-person care.[4] Additionally, remote technology has proved surprisingly popular with both

**BMJ**

clinicians and patients.[3] However, some patients are likely to find remote technology less helpful than others. They may lack necessary equipment and reliable internet connectivity, be unfamiliar and uncomfortable with the technology, or have mental disorders such as dementia[5] and schizophrenia[6] that affect their motivation and cognitive functioning making it difficult to participate in remote care. Patients, carers and clinicians may also have concerns about privacy and security.[7] The potential value of remote technology in mental healthcare has yet to be systematically evaluated and the evidence base to support best practice is sparse. The potential benefits of remote care must be balanced against the risk that its use might, in some situations, have adverse effects on quality of care, patient satisfaction and clinical outcomes. There is a pressing need to formally evaluate the impact of remote care technology on mental healthcare so that potential barriers to accessing remote care can be identified, its benefits maximised and potential risks minimised.

Electronic health record (EHR) data could be used to address this gap in understanding. EHRs have been widely adopted in National Health Service (NHS) healthcare services.[8] Information is stored as both structured data, which includes categorical data such as demographics, prescribed medications and diagnoses, and, to a greater extent, unstructured data such as free text notes which tend to be written in narrative format by clinical or administrative staff. Unstructured data may contain details on symptoms, diagnoses and treatment plans that are otherwise not available in structured data fields. EHRs support individual patient care and have the potential to support largescale research through the analysis of deidentified clinical data.[9] Natural language processing (NLP) techniques can be used to automatically and rapidly extract clinical information from unstructured free text clinical notes and correspondence to assess the associations of clinical factors,[9 10] environmental factors[11] and treatments[12 13] with clinical outcomes.[14]

We aim to investigate the prevalence and distribution of remote assessments to better understand variation in uptake based on clinical and demographic characteristics and to investigate differences in clinical outcomes associated with remote versus in-person assessments using a large sample of patients receiving care for mental disorders by applying NLP to extract data from deidentified EHR records.

The rationale for conducting this study is to contribute to the evidence base related to the use of remote mental healthcare that can positively impact clinical practice and service delivery. The ability to support greater access to mental healthcare through remote technology would reduce barriers and costs in providing care to people who would benefit from this approach while increasing capacity to provide in-person care to those who would prefer or cannot be safely treated remotely. This could enable more patients to receive timely care and improve clinical outcomes, quality of life and functioning and improve cost-effectiveness of mental healthcare provision.

## METHODS AND ANALYSIS
### Study aims and design
We will conduct two studies. First, we aim to conduct a retrospective, longitudinal cohort study to examine the potential associations between remote care and clinical outcomes and to test the hypothesis that (a) use of remote care will be lower in patients with poor motivation compared with in-person care and (b) remote care use will be more frequent in the treatment of less severe disorders, such as mild depression and anxiety, whereas in-person care will be more frequent in the treatment of more severe disorders, such as psychotic and bipolar mood disorders.

Second, an NLP study will compare clinical documentation between in-person appointments and remote care appointments. We hypothesise that (a) the documentation of diagnosis, symptoms and treatments will be similar for remote and in-person care, (b) documentation of clinical signs (as part of mental state examination) will be less frequent with remote than with in-person care and (c) use of remote care will be associated with fewer documented episodes of suicidal thoughts. We hypothesise that differences between remote and in-person care will be less evident when remote assessment involves video, as opposed to audio or text.

### Study setting
The data used in this study will be obtained from the South London and Maudsley (SLaM) NHS Foundation Trust. SLaM is one of the largest providers of specialist mental healthcare in Europe and provides more than 230 services, including specialist psychosis services, inpatient wards, community and outpatient services, within its catchment serving approximately 1.3 million residents in the London boroughs of Lambeth, Southwark, Lewisham and Croydon. SLaM treats over 5000 inpatients and 40 000 outpatients per year. The Trust has implemented a full EHR system since 1 April 2006 and the SLaM NIHR Biomedical Research Centre (BRC) Case Register contains the clinical records of 417 482 patients (accurate on 08/02/2022).[15]

### Data source
Since April 2006, SLaM has used an electronic psychiatric clinical records system, the electronic Patient Journey System (ePJS) to record instances of patients accessing or interacting with mental healthcare services. The SLaM BRC Case Register comprises deidentified clinical data from ePJS including structured fields (for demographic, diagnostic and medication data) and deidentified unstructured free text fields from case notes and correspondence. Unstructured clinical text is documented by healthcare professionals during the provision of mental healthcare to patients and includes history, mental state examination, diagnostic formulation and care plans. Healthcare professionals who document clinical data include psychiatrists, psychologists, nursing staff, care coordinators and allied healthcare professionals. A patient-led oversight committee considers all proposed research before access to the deidentified data is permitted and access to data

is restricted to honorary or substantive employees of SLaM.[16]

Data for this study will be obtained from the SLaM BRC Case Register using the Clinical Record Interactive Search tool (CRIS), a bespoke Microsoft Structured Query Language (SQL) database query and assembly tool that allows authorised researchers to extract and recode data from the SLaM BRC Case Register. Data are obtained from structured EHR fields and from unstructured free-text documentation using NLP.[8] Data analysis will start on 1 February 2023.

## Natural language processing

NLP is an information extraction technique used to automatically identify relevant information from unstructured free text. NLP models are developed by defining a construct of interest (eg, a particular clinical symptom or outcome), assembling a corpus of documents potentially containing the construct of interest (eg, a selection of deidentified clinical documents from EHRs) and manually annotating examples from the corpus to classify true positive instances of the construct of interest. The annotated examples may be used as training and reference data to develop and evaluate the accuracy of a machine learning-derived NLP algorithm to automatically identify the presence of the construct in other documents.

To maximise ascertainment of clinical data in this study, as well as assembling structured data, NLP models which have previously been developed to extract mental disorder symptom data from EHR documents will be applied.[17] These NLP models have been developed to enrich the SLaM BRC Case Register by allowing clinical information to be extracted from unstructured free text which comprise assessments, reviews and correspondence documented by mental healthcare professionals. The data extracted by NLP include a range of symptoms, contextual factors (certain physical health disorders, exposure to illicit substances and social factors), interventions (medications and psychological therapy) and clinical outcomes.[17 18]

The performance of an NLP model is measured through precision and recall. Precision refers to the number of true positive instances that the NLP model has identified, divided by the total number of instances retrieved by the NLP model (both true positive and false positive instances). Recall refers to the number of true positive instances identified by the NLP model, divided by the total number of existing true positives in the entire sample. Precision and recall statistics for the NLP models employed in this project are supplied in online supplemental table 1. The full library of NLP models available within CRIS and details of their development are provided on the SLaM CRIS website.[18]

NLP-derived data will be ascertained as binary variables which indicate the presence of at least one mention of the construct of interest in each analysed document.

## Study population

Data will be extracted for all patients of all ages who had an active episode of care from 1 January 2019 to 31 March 2022. An active episode of care is defined as at least one clinical contact with a healthcare professional with SLaM services during the study period.

## Index dates

Data extracted for both studies will be from between 1 January 2019 and 31 March 2022. This time-period was selected to provide data on clinical assessments prior to and during the COVID-19 pandemic (declared by the World Health Organisation in March 2020)[19] and on follow-up data relating to longitudinal clinical outcomes.

The index date for the cohort study will be 1 January 2021. For the longitudinal cohort study, all patients who have an active episode of care during 2019 and during 2020 will be considered. Follow-up data will be obtained from 1 January 2021 to 31 March 2022. For the NLP study, data on clinical assessments and clinical outcomes will be considered from 1 January 2019 to 31 March 2022.

## Data extraction

Data derived from the SLaM BRC Case register will be interrogated using SQL, a standardised language for accessing and manipulating relational databases. SQL enables the application of queries to gather data from different parts of the SLaM BRC Case Register at the individual patient level. The use of SQL with EHR databases allows researchers to identify specific patient samples and extract relevant longitudinal clinical data. Demographic data on age, gender, ethnicity, marital status, occupation, housing status, multiple deprivation score and date of death and clinical data on psychiatric disorder diagnosis and comorbid diagnoses will be extracted using SQL.

Data on type of clinical event attended (in-person versus remote including did not attend (DNA)), type of remote event (video, audio or text), number of hospital admissions (voluntary and involuntary under the UK Mental Health Act), date of hospital admission, date of discharge, number of A&E Liaison Psychiatry care episodes, number of Crisis Resolution Home Treatment Team (CRHTT) care episodes and whether psychotropic medications (antidepressants, antipsychotics and mood stabilisers) were prescribed will be extracted to support descriptive and inferential analyses.

## Patient and public involvement

The study was developed in collaboration with the SLaM CRIS Service User and Carer Advisory Group, which includes patients and carers who have experience of SLaM mental healthcare and experience of supporting EHR data research studies. The study has received approval from the SLaM CRIS Oversight Committee (chaired by a patient representative) which reviews and approves applications to extract and analyse the data. The CRIS infrastructure was designed and is managed with ongoing patient and carer input to ensure all research projects comply with data governance, legal and ethical guidelines. Further details are available on the SLaM CRIS website.[20]

## Defined variables
### Predictors
The type of clinical event attended by a patient, categorised as in-person, remote (video call, phone, letter, instant or short message service text message, email) or DNA. Inpatient clinical events will be excluded (as they are conducted in-person) and only data from community clinical events will be analysed. A composite predictor variable will be generated as the proportion of all appointments (denominator) that are remote care appointments (numerator) between 1 January 2019 and 31 December 2020 as a measure of exposure to remote care. Also, a variable indicating the presence of documented poor motivation will be generated from unstructured text using NLP. To take into account potential differences in the use of remote care during periods of travel restrictions during the COVID-19 pandemic, supplementary analyses will be conducted for the period between 1 April 2020 and 30 June 2021.

### Covariates
Age, gender, ethnicity, occupation, marital status, psychiatric disorder ICD-10 diagnosis (categorised according to major disorder groups defined in a previous study),[21] housing status and index of social deprivation (obtained through a data linkage with the London Data Store[22] at Lower Super Output Area level). For patients in the cohort with at least one clinical event documented on or prior to 31 December 2019, a supplementary analysis will be conducted (provided a sufficient sample size is available) in which the number of outpatient appointments and number of days spent in a psychiatric hospital during 2020 will be included as a covariate to adjust for the potential association of prior healthcare service utilisation with clinical outcomes.

Covariate categories will be coded according to NHS Data Dictionary definitions.[23] All predictor and covariate variables will be obtained from structured EHR data fields except for psychiatric disorder diagnosis, marital status and housing status which will be obtained from both structured and unstructured data using NLP using methods described subsequently.

### Outcome measures
#### Cohort study
Primary outcome variable: psychiatric hospital admissions categorised as voluntary or involuntary (under a section of the UK Mental Health Act). Psychiatric hospital admission was chosen as the primary outcome variable to compare illness severity associated with the type of clinical event attended by a patient.

Secondary outcome variables: A&E Liaison Psychiatry care episodes, CRHTT care episodes, psychotropic medication prescribing (antidepressants, antipsychotics and mood stabilisers defined according to the British National Formulary,[24] ICD-10 psychiatric disorder diagnosis, attendance/non-attendance of in-person appointments, attendance/non-attendance of remote care appointments and mortality as recorded in EHR data and through a pre-existing data linkage with the Office for National Statistics to obtain data on cause of death including natural causes, suicide or open coroners' verdict or other unnatural causes (unlawful killing, accident or misadventure).

All outcome variables will be obtained from structured EHR data fields.

#### NLP study
Outcome variables: difference in frequency in documentation of diagnosis, symptoms, clinical signs (as part of mental state examination), treatments and episodes of suicidal thoughts between clinical documents related to in-person appointments compared with documents related to remote care appointments.

## Statistical analysis
Data analysis will be performed using R statistical computing and graphics software.

### Cohort study
#### Descriptive statistics
Descriptive statistics will be obtained for demographic variables, including age, gender, ethnicity, marital status, housing status, occupation and multiple deprivation score. Frequencies and percentages will be obtained for categorical variables and mean and SD will be obtained for continuous variables.

#### Multivariable regression analyses
Multivariable linear regression will be used to analyse the potential association between type of clinical event attended and number of days spent in a psychiatric hospital during the follow-up period,

Multivariable logistic regression will be used to analyse the potential associations between type of clinical event attended and any documentation of (i) poor motivation; (ii) primary psychiatric disorder diagnosis; (iii) antidepressants; (iv) antipsychotics and (v) mood stabilisers during the follow-up period.

Multivariable Cox regression will be used to analyse the potential associations between type of clinical event attended and time to (i) voluntary psychiatric hospital admission; (ii) involuntary psychiatric hospital admission; (iii) A&E Liaison Psychiatry care episode; (iv) CRHTT care episode and (v) mortality.

### NLP Study
NLP will be used to obtain data using applications previously developed by RP in collaboration with the NIHR Maudsley BRC.[9 10 25 26] Online supplemental table 1 summarises the NLP applications and full details of the specification and precision statistics are available from the CRIS NLP Library.[18]

NLP data will be obtained for each unstructured document representing an in-person or remote care event. The number of in-person versus remote care documents containing each NLP-derived construct will be compared using $\chi^2$ tests and logistic regression (predictor variable: type of clinical contact; outcome variable: presence of NLP-derived construct).

## Sample size

As sample sizes of number of patients and number of documents in the SLaM BRC Case Register can be estimated from previous years' data, the pwr package (R V.4.0.2) was used to estimate the minimal detectable effect size (MDE) for the cohort study and NLP study analyses using a significance level of 0.001 and power of 0.8 as follows:

Cohort study: MDE Cohen's w (sample size: 78 805 active patients in 2018 and 2019; df: 20): 0.022.

NLP Study: MDE Cohen's w (sample size: 5 083 863 documents in 2018 and 2019; df: 100): 0.004.

This indicates the data available will be well powered to detect small changes in effect size.

## Missing data

Based on previous studies, data on the predictor variable, age, gender and all outcome variables will be complete within the SLaM BRC Case Register. Where there are missing data for other covariates, two analyses will be conducted: a full sample analysis including missing data as a category in regression analyses and a complete case analysis dropping individuals who have one or more missing covariates.

## Ethics and dissemination

The SLaM BRC Case Register and CRIS tool have received ethical approval as a deidentified dataset for secondary mental health research analyses from Oxfordshire REC C (Ref: 18/SC/0372). The study has received approval from the SLaM CRIS Oversight Committee (chaired by a patient representative) which reviews and approves applications to extract and analyse the data. The CRIS infrastructure was designed and is managed with ongoing patient and carer input to ensure all research projects comply with data governance, legal and ethical guidelines. Further details are available on the SLaM CRIS website.[20] Study findings will be disseminated through peer-reviewed journal publications and published open access to ensure there are no limitations to readership and access to their content are as widely available as possible.

## DISCUSSION
### Strengths

This study aims to investigate the associations of remote mental healthcare with clinical outcomes by analysing structured and unstructured data from a large, deidentified EHR dataset. The study is strengthened by the use of routinely recorded data that are representative of clinical practice. Thus, the findings of the study will be generalisable to patients receiving care for a wide range of mental disorders and across a broad range of sociodemographic characteristics. The study will provide key insights into the factors associated with higher or lower rates of remote care appointments and provide insights for healthcare policymakers to reduce barriers to access remote care and contribute to evidence-based guidelines for best practice.

The availability of unstructured EHR data will enable the assembly and analysis of rich clinical symptom data through the use of NLP to better understand how remote care appointments vary from in-person appointments in terms of the nature of clinical information documented by mental healthcare professionals. This information could help better understand the potential benefits and limitations of remote care based on clinical presentation and treatment needs.

## Limitations

As this study is observational and based on data that have already been recorded as part of routine healthcare delivery, it will not be possible to determine or infer causal associations between exposure to remote care and clinical outcomes. This would require a prospectively conducted study. It is also possible that our findings could be confounded by clinical indication. For example, clinicians may perceive a patient with poor motivation to be less likely to engage with remote care and may preferentially offer in-person appointments to such individuals.

The dataset analysed in the study will comprise EHR data from secondary mental healthcare services. Data from primary care services will not be available. This means that it will not be possible to draw inferences on the use of remote care for mental disorders in primary care. Furthermore, data will only be available for patients receiving care from the SLaM NHS Trust. It will not be possible to ascertain predictor, covariate or clinical outcome data for patients who are discharged during the period of the study (eg, due to moving outside of the SLaM catchment area).

A further limitation is the presence of missing data. As EHR data are recorded as part of routine clinical practice, data are not systematically ascertained on every potential clinical symptom or feature a patient may or may not present with. This means that there may be missing symptom data which are not documented by clinicians and so would not be ascertained by NLP models. Nonetheless, it is likely that a clinically relevant or significant feature of a patient's presentation would be documented in the EHR if it contributes towards the nature and/or degree of the mental disorder and/or decisions related to treatment planning.

## Data management and confidentiality

To comply with all necessary data protection regulations, EHR data will be stored and analysed using encrypted workstations and virtual machines hosted in a UK-based Microsoft Azure datacentre.

**Author affiliations**
[1]Department of Psychosis Studies, Division of Academic Psychiatry, Institute of Psychiatry Psychology and Neuroscience, London, UK
[2]NIHR Maudsley Biomedical Research Centre, South London and Maudsley Mental Health NHS Trust, London, UK
[3]Department of Psychiatry, University of Oxford, Oxford, UK
[4]NIHR Oxford Health Biomedical Research Centre, Oxford, UK
[5]Department of Brain and Behavioral Sciences, University of Pavia, Pavia, Italy
[6]Department of Psychological Medicine, Division of Academic Psychiatry, Institute of Psychiatry Psychology and Neuroscience, London, UK

**Contributors** The studies were conceived and designed by RP. Protocol preparation, literature review, statistical analysis plan and manuscript preparation was performed by MSA, DK, DO, PF-P and RP. All authors approved the final manuscript.

**Funding** RP has received funding from an NIHR Advanced Fellowship (NIHR301690), a Medical Research Council (MRC) Health Data Research UK Fellowship (MR/S003118/1) and an Academy of Medical Sciences Starter Grant for Clinical Lecturers (SGL015/1020) supported by the Wellcome Trust, MRC, British Heart Foundation, Arthritis Research UK, the Royal College of Physicians and Diabetes UK.

**Disclaimer** This project is funded by the National Institute for Health Research (Advanced Fellowship), grant number NIHR301690. The views expressed are those of the authors and not necessarily those of the NHS, the NIHR or the Department of Health. The funders had no role in the design and conduct of the study; collection, management, analysis and interpretation of the data; preparation, review or approval of the manuscript and decision to submit the manuscript for publication.

**Competing interests** RP has received funding from Janssen and Holmusk, outside the submitted work. The other authors declare no competing interests.

**Patient and public involvement** Patients and/or the public were involved in the design, or conduct, or reporting, or dissemination plans of this research. Refer to the Methods section for further details.

**Patient consent for publication** Not applicable.

**Ethics approval** This study involves data from human participants and was approved by Oxfordshire REC C (Ref: 18/SC/0372). The CRIS data resource received ethical approval as a deidentified dataset for secondary mental health research analyses from Oxfordshire REC C (Ref: 18/SC/0372). The study was approved by the SLaM BRC CRIS oversight committee. Consent is not required to analyse the deidentified dataset for approved research studies. Patients may opt out of inclusion in the deidentified dataset.

**Provenance and peer review** Not commissioned; externally peer reviewed.

**Data availability statement** No data are available. No additional data are available.

**ORCID iDs**
Muhammad Shamim Ahmed http://orcid.org/0000-0002-9260-8958
Dominic Oliver http://orcid.org/0000-0002-8920-3407
Paolo Fusar-Poli http://orcid.org/0000-0003-3582-6788
Rashmi Patel http://orcid.org/0000-0002-9259-8788

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
