## [Reviewer comments · BMJ Open]

ARTICLE DETAILS

TITLE (PROVISIONAL)	Associations of remote mental healthcare with clinical outcomes: a natural language processing enriched electronic health record data study protocol.
AUTHORS	Ahmed, Muhammad; Kornblum, Daisy; Oliver, Dominic; Fusar-Poli, Paolo; Patel, Rashmi

VERSION 1 – REVIEW

REVIEWER	Weich, Scott University of Sheffield, School of Health and Related Research
REVIEW RETURNED	31-Aug-2022

GENERAL COMMENTS	This is a really interesting study, and I can't wait to see the results. I am concerned, however, that there are one or two limitations that the authors have't fully addressed. It may be too late to alter the design of the research but it would good if these could be addressed in the manuscript. In particular, I am concerned about the possibility (indeed likelihood) of confounding by indication. In other words, contacts that are urgent, where patients are acutely unwell and where risk is increased are much (much) more likely to be seen in person. This includes crisis team contacts, liaison reviews, community home visits and mental health act assessments. These contacts are the most likely to result in further unplanned care, including admission and other restrictions. The authors make no mention of how they will handle contacts in inpatient settings, though perhaps these will be excluded. I note also that the list of covariates is quite limited and does not include previous admissions or detentions. I am also concerned about the potential for bias by, for example, gender, ethnicity, language and age. Those of us unfamiliar with NLP would benefit from an account of how this will be addressed. There are one or two other minor issues that might be addressed: 1) There is a formatting error at the top of p9 which means that the first line is incomplete, as is the beginning of the second paragraph.2) I would like to know how the authors plan to define, operationalise and measure 'poor motivation'.
--

REVIEWER	Edgcomb , J University of California Los Angeles, Department of Psychiatry and Behavioral Sciences
REVIEW RETURNED	18-Nov-2022

GENERAL COMMENTS

This is a study protocol for two studies leveraging EHR data with NLP methods to examine associations of remote (vs. in-person) mental health care with clinical outcomes. The authors should be commended for a timely and well-written protocol that is of high clinical relevance. Here are several points of clarification:

[1] Abstract:

- (1) Include brief specification of the NLP techniques proposed in the methods and analysis section
- (2) Regarding Ethics and Dissemination, specification of approval of use of narrative text records (including identifiers) in the second sentence would add clarity.

[2] Introduction:

Given study aim (a) (page 4, lines 53-54) focuses on associations of clinical characteristics with uptake of remote care, the stated overarching aim (page 4, lines 41-44) could be clarified by specifying not only the differences in clinical outcomes but also differences in remote mental health uptake/use.

[3] Study aims and design:

- (1) Consider strengthening the justification for the hypothesis (page 4, line 53) that remote care use will be lower in patients with poor motivation. Motivation is mentioned once (page 4, line 19), and some further detail on existing research in this domain would bolster this hypothesis.
- (2) Also, clarifying (page 4, line 53) that the study compares remote care use with in-person care use (e.g., "use of remote care, compared with in-person care, will be..."), rather than compares remote care use with no care use.
- (3) Regarding "use of remote care will be associated with fewer documented episodes of suicidal thoughts" (page 5, line 4): please clarify whether it is anticipated remote care (vs. in-person care) will be associated with fewer documented affirmations or negations (or any documentation) of suicidal thoughts? In other words, it is anticipated suicidal thoughts are more prevalent among individuals receiving in-person (vs. remote care)? Or is it anticipated that there is greater documentation of suicide risk assessment (including positive and negative screening for suicidal thoughts) among individuals receiving in-person care?
- (4) Also (page 5, line 4), consider including justification for the selection of suicidal thoughts rather than the broader range of self-injurious thoughts and behaviors (including suicide attempts, preparatory acts, and non-suicidal self-injury)
- (5) Clarify (page 5, line 5-6) whether the hypothesis that "differences between remote and in-person care will be less evident when remote assessment involves video, as opposed to audio or text" applies to the first study as well as the second.

[4] Natural Language Processing:

- (1) Page 5, line 52: Typo? "Their performance of an NLP model..."
- (2) Handling of affirmations, negations, and contextual factors related to NLP would be helpful to describe. One example: regarding aggression - a patient could be documented to be presently aggressive with nursing staff (affirmed), then calm (negated), then described as aggressive but only in the past (affirmed, past), then describes being aggressed upon (e.g., hit by another patient) (affirmed, referring to another individual), then describes homicidal ideation without aggressive behavior (ambiguous). Does the NLP-enrichment have capacity to navigate

these nuances? In other words, further description of the limits and details of the NLP methods would additionally strengthen the protocol.

[5] Study Population:

(1) Please define "at least one clinical contact with a healthcare professional" - what types of encounters will this include? e.g., page 5, line 5-6 includes text-based encounters - will clinical contact include e.g., text messages?

(2) Consider including the anticipated demographic breakdown of the study population (e.g., the percentage of population served that is ≤ 18 years old)

[6] Defined Variables - please clarify how the following will be ascertained using EHR data elements:

(1) Motivation (page 7, line 14) - further information on how documentation of poor motivation will be used as a data element would be helpful: Will this refer to documented poor motivation at index encounter? At previous encounters?

*Specifically, my concern here is that the hypothesis is "use of remote care will be lower in patients with poor motivation". If motivation is operationalized as provider documentation of poor motivation at that encounter, then perhaps it is not that use of remote care will be lower in patients with poor motivation, but rather use of remote care will result in more frequent documentation of poor motivation. Some comment on how the study team plans to address this confounding of remote vs in-person documentation effects with variables of interest would be very helpful.

(2) Severity of illness (page 4, line 54) - please clarify how severity will be defined (ICD code alone vs. in combination with other data elements such as number of hospitalizations)

(3) ICD-10 diagnoses (page 7, line 31) - How will ICD code groups be categorized? Will a standardized system be used? Sidenote: might consider possible inclusion of ICD code-based score for medical comorbidity (e.g., Charlson, Elixhauser)? If including children, consider pediatric mental health specific ICD code categorization (e.g., <https://pubmed.ncbi.nlm.nih.gov/32202603/>) to capture child/adolescent disorders?

(4) Medications (page 7, line 30) - Will a structured system be used (e.g. RxNorm, ATC)?

(5) Suicidal thoughts - see comment above re page 5, line 4

[7] Multivariable Logistic regression

(1) Page 8, line 3-12 - pasted text? This seems to be text for another section

[8] Missing data

(1) Additional detail on handling of missing data would be helpful. Specifically: please clarify how missing data for NLP terms will be handled. There are many unstructured data elements that one could anticipate will be highly missing (e.g., whether motivation or suicidal thoughts etc. were documented in a given clinical note). These data are likely missing not at random (e.g., contingent upon diagnosis, clinical presentation, type of provider seen and type of clinical note) and may be quite sparse.

General comments:

[1] If the study will include children and adolescents, please indicate how (if at all) these data will be considered separately from

	adults. For example, how will affirmation of motivation by a parent and negated motivation by the child (patient) will be handled? [2] How will the COVID-19 pandemic and associated shifts be handled in the data analysis? Page 6, lines 17-20 describe the timeline including a longitudinal cohort study through the end of 2020. Will analyses account for shutdown/closures occurring during the Spring of 2020?
--	--

VERSION 1 – AUTHOR RESPONSE

Reviewer: 1

Prof. Scott Weich, University of Sheffield

Comments to the Author:

This is a really interesting study, and I can't wait to see the results.

I am concerned, however, that there are one or two limitations that the authors have't fully addressed. It may be too late to alter the design of the research but it would good if these could be addressed in the manuscript.

/*Thank you for your interest in our study and supportive comments.*/*

In particular, I am concerned about the possibility (indeed likelihood) of confounding by indication. In other words, contacts that are urgent, where patients are acutely unwell and where risk is increased are much (much) more likely to be seen in person. This includes crisis team contacts, liaison reviews, community home visits and mental health act assessments. These contacts are the most likely to result in further unplanned care, including admission and other restrictions. The authors make no mention of how they will handle contacts in inpatient settings, though perhaps these will be excluded.

/*We agree that there is the potential for confounding by indication with respect to the decision of whether to offer a remote or in-person clinical appointment based on the nature and/or degree of illness severity. Indeed, a key aim of the study is to better understand the factors which are associated with exposure to remote mental healthcare. A previous study (Patel R, Irving J, Brinn A, et al. Impact of the COVID-19 pandemic on remote mental healthcare and prescribing in psychiatry: An electronic health record study. *BMJ Open*. 2021;11(3):e046365. doi:10.1136/bmjopen-2020-046365) has already demonstrated variation by age group. The proposed study will seek to address the degree to which other clinical factors outlined in *Defined variables* section of the protocol (Page 7) are associated with a high or low proportion of remote care appointments. As the study is observational and based on already recorded data, it will not be possible to determine or infer causal associations between remote care appointments and clinical outcomes and this would require a prospective study. We have updated the protocol (Page 10, Paragraph 3) to clarify this point. As you rightly point out, inpatient contacts are conducted in person and so will be excluded in the data assembly/analysis. We have updated the protocol (Page 7, Paragraph 4) to clarify this point.*/*

I note also that the list of covariates is quite limited and does not include previous admissions or detentions.

/*We agree that previous healthcare service utilisation (including previous voluntary and involuntary psychiatric hospital admissions) may be associated with subsequent clinical outcomes. However, only a subset of the cohort will have data on prior healthcare service utilisation. We propose to conduct a supplementary analysis on the cohort of patients with an active episode of care during 2019/20 who have at least 12 months of prior data available in the dataset (i.e. at least one clinical encounter on or

prior to 31st December 2017). If a sufficient sample size is available, we will include the number of appointments and number of days spent in hospital during 2018 as a covariate in this supplementary analysis in order to adjust for the potential associations of prior healthcare service utilisation with clinical outcomes. We have updated the protocol (Page 7, Paragraph 5) to describe this supplementary analysis.*/

I am also concerned about the potential for bias by, for example, gender, ethnicity, language and age.

/*We agree that these are potentially confounding variables. Where available (age, gender and ethnicity), these variables will be included as covariates in multivariable regression analysis as described in the Covariates section of the protocol (Page 7). Unfortunately, data on primary spoken language is not readily accessible in the dataset and so cannot be analysed.*/

Those of us unfamiliar with NLP would benefit from an account of how this will be addressed.

/*We have updated the section on natural language processing (Page 5, Paragraph 6) to provide further information.*/

There are one or two other minor issues that might be addressed:

1) There is a formatting error at the top of p9 which means that the first line is incomplete, as is the beginning of the second paragraph.

/*Thank you for spotting this. We have now corrected this section.*/

2) I would like to know how the authors plan to define, operationalise and measure 'poor motivation'.

/*Poor motivation will be ascertained through the application of an NLP model on unstructured EHR documents as described in the *Predictors* section of the protocol (Page 7).*/

Reviewer: 2

Dr. J Edgcomb , University of California Los Angeles

Comments to the Author:

This is a study protocol for two studies leveraging EHR data with NLP methods to examine associations of remote (vs. in-person) mental health care with clinical outcomes. The authors should be commended for a timely and well-written protocol that is of high clinical relevance. Here are several points of clarification:

/*Thank you for your supportive comments.*/

[1] Abstract:

(1) Include brief specification of the NLP techniques proposed in the methods and analysis section

/*We have added some additional wording to the abstract to provide additional context.*/

(2) Regarding Ethics and Dissemination, specification of approval of use of narrative text records (including identifiers) in the second sentence would add clarity.

/*We have added this information to the abstract. Please note that the ethical approval does not include access to analyse patient identifiers and it is only possible to analyse de-identified data (as stated in the protocol).*/

[2] Introduction:

Given study aim (a) (page 4, lines 53-54) focuses on associations of clinical characteristics with uptake of remote care, the stated overarching aim (page 4, lines 41-44) could be clarified by specifying not only the differences in clinical outcomes but also differences in remote mental health uptake/use.

*/*We have updated the introduction to include this aim (Page 4, Paragraph 4).*/*

[3] Study aims and design:

(1) Consider strengthening the justification for the hypothesis (page 4, line 53) that remote care use will be lower in patients with poor motivation. Motivation is mentioned once (page 4, line 19), and some further detail on existing research in this domain would bolster this hypothesis.

*/*We have updated the introduction to include further references related to the potential impact of problems with motivation, mental disorders and remote care technology (Page 4, Paragraph 2).*/*

(2) Also, clarifying (page 4, line 53) that the study compares remote care use with in-person care use (e.g., "use of remote care, compared with in-person care, will be..."), rather than compares remote care use with no care use.

*/*We have updated the study aims to clarify this point (Page 5, Paragraph 1).*/*

(3) Regarding "use of remote care will be associated with fewer documented episodes of suicidal thoughts" (page 5, line 4): please clarify whether it is anticipated remote care (vs. in-person care) will be associated with fewer documented affirmations or negations (or any documentation) of suicidal thoughts? In other words, it is anticipated suicidal thoughts are more prevalent among individuals receiving in-person (vs. remote care)? Or is it anticipated that there is greater documentation of suicide risk assessment (including positive and negative screening for suicidal thoughts) among individuals receiving in-person care?

*/*It is anticipated that remote care will be associated with fewer documented episodes of suicidal thoughts compared to in-person care.*/*

(4) Also (page 5, line 4), consider including justification for the selection of suicidal thoughts rather than the broader range of self-injurious thoughts and behaviors (including suicide attempts, preparatory acts, and non-suicidal self-injury)

*/*We have chosen to investigate suicidal thoughts based on the availability of an NLP model to ascertain suicidal thoughts in unstructured EHR data. Unfortunately an NLP model to ascertain self-injurious thoughts and behaviours is not currently available for the dataset being analysed in this study.*/*

(5) Clarify (page 5, line 5-6) whether the hypothesis that "differences between remote and in-person care will be less evident when remote assessment involves video, as opposed to audio or text" applies to the first study as well as the second.

*/*This hypothesis only applies to the second study investigating clinical documentation.*/*

[4] Natural Language Processing:

(1) Page 5, line 52: Typo? "Their performance of an NLP model..."

*/*Thank you for pointing this out. We have corrected this error.*/*

(2) Handling of affirmations, negations, and contextual factors related to NLP would be helpful to describe. One example: regarding aggression - a patient could be documented to be presently aggressive with nursing staff (affirmed), then calm (negated), then described as aggressive but only in the past (affirmed, past), then describes being aggressed upon (e.g., hit by another patient) (affirmed, referring to another individual), then describes homicidal ideation without aggressive behavior (ambiguous). Does the NLP-enrichment have capacity to navigate these nuances? In other words, further description of the limits and details of the NLP methods would additionally strengthen the protocol.

/*We agree, it would be interesting to investigate the valence of NLP-derived data. Unfortunately this is beyond the scope of the present study as the NLP models available are designed to ascertain positive instances of documented symptoms.*/*

[5] Study Population:

(1) Please define "at least one clinical contact with a healthcare professional" - what types of encounters will this include? e.g., page 5, line 5-6 includes text-based encounters - will clinical contact include e.g., text messages?

/*The types of encounter are defined in the *Predictors* section of the protocol (Page 7).*/*

(2) Consider including the anticipated demographic breakdown of the study population (e.g., the percentage of population served that is ≤ 18 years old)

/*We will obtain data on demographic characteristics of the population listed under the *Covariates* section of the protocol (Page 7). Descriptive statistics will be obtained for these variables (Page 8).*/*

[6] Defined Variables - please clarify how the following will be ascertained using EHR data elements:

(1) Motivation (page 7, line 14) - further information on how documentation of poor motivation will be used as a data element would be helpful: Will this refer to documented poor motivation at index encounter? At previous encounters?

/* Poor motivation will be ascertained through the application of an NLP model on unstructured EHR documents as described in the *Predictors* section of the protocol (Page 7). NLP-derived data will be ascertained as binary variables with "1" indicating at least one mention of the construct of interest in the document being analysed (Page 6, Paragraph 4).*/*

*Specifically, my concern here is that the hypothesis is "use of remote care will be lower in patients with poor motivation". If motivation is operationalized as provider documentation of poor motivation at that encounter, then perhaps it is not that use of remote care will be lower in patients with poor motivation, but rather use of remote care will result in more frequent documentation of poor motivation. Some comment on how the study team plans to address this confounding of remote vs in-person documentation effects with variables of interest would be very helpful.

/*We agree, it is indeed possible that the data may be confounded on the basis that clinicians may choose to offer in-person appointments to patients they consider to be less likely to engage with remote care appointments due to poor motivation. We have updated the limitations section of the discussion to highlight this (Page 10, Paragraph 3).*/*

(2) Severity of illness (page 4, line 54) - please clarify how severity will be defined (ICD code alone vs. in combination with other data elements such as number of hospitalizations)

/*Severity will be defined by whether patients are admitted to a psychiatric hospital voluntarily or involuntarily under a section of the UK Mental Health Act (Page 7/8).*/

(3) ICD-10 diagnoses (page 7, line 31) - How will ICD code groups be categorized? Will a standardized system be used? Sidenote: might consider possible inclusion of ICD code-based score for medical comorbidity (e.g., Charlson, Elixhauser)? If including children, consider pediatric mental health specific ICD code categorization (e.g., <https://pubmed.ncbi.nlm.nih.gov/32202603/>) to capture child/adolescent disorders?

/*ICD-10 diagnosis will be categorised according to major disorder groups as defined by a previous study on the same dataset (<https://pubmed.ncbi.nlm.nih.gov/28355424/> - eTable 1). We have updated the *Covariates* section accordingly (Page 7)*/

(4) Medications (page 7, line 30) - Will a structured system be used (e.g. RxNorm, ATC)?

/*Medication data will be obtained according to the categories of the British National Formulary. We have updated the Outcome measures section accordingly (Page 8).*/

(5) Suicidal thoughts - see comment above re page 5, line 4

/*Please see previous response.*/

[7] Multivariable Logistic regression

(1) Page 8, line 3-12 - pasted text? This seems to be text for another section

/*Thank you for spotting this. We have amended this error.*/

[8] Missing data

(1) Additional detail on handling of missing data would be helpful. Specifically: please clarify how missing data for NLP terms will be handled. There are many unstructured data elements that one could anticipate will be highly missing (e.g., whether motivation or suicidal thoughts etc. were documented in a given clinical note). These data are likely missing not at random (e.g., contingent upon diagnosis, clinical presentation, type of provider seen and type of clinical note) and may be quite sparse.

/*NLP-derived data will be obtained as a binary variable indicating the presence of at least one mention of a construct of interest within a document. Where there is no mention of a construct of interest, this will be considered to be the absence of any documentation related to that construct. As this is a study undertaken on already recorded data, it is not possible to ascertain the degree to which clinical constructs are not documented when they were actually present. We have updated the limitations section of the manuscript to highlight this (Page 10, Paragraph 5).*/

General comments:

[1] If the study will include children and adolescents, please indicate how (if at all) these data will be considered separately from adults. For example, how will affirmation of motivation by a parent and negated motivation by the child (patient) will be handled?

/*It is not possible to determine affirmation of motivation by a parent vs a child (patient) with the NLP models available for the analysis of this dataset. It will therefore not be possible to undertake this analysis.*/

[2] How will the COVID-19 pandemic and associated shifts be handled in the data analysis? Page 6, lines 17-20 describe the timeline including a longitudinal cohort study through the end of 2020. Will analyses account for shutdown/closures occurring during the Spring of 2020?

/*Thank you for raising this point. We agree that the use of remote care during periods of travel restrictions may contribute to variation in the data being analysed. We therefore plan to conduct a supplementary analysis for the period between 1st March 2020 and 31st July 2021 which encompass the principle periods of national COVID-19 restrictions in the UK (Page 7, Paragraph 4).*/

Reviewer: 1

Competing interests of Reviewer: None

Reviewer: 2

Competing interests of Reviewer: I have research funding related to clinical informatics and natural language processing from the National Institute of Health, Brain and Behavior Research Foundation, Thrasher Research Fund, and Sorensen Foundation.

VERSION 2 – REVIEW

REVIEWER	Weich, Scott University of Sheffield, School of Health and Related Research
REVIEW RETURNED	01-Feb-2023
GENERAL COMMENTS	The authors have done a very good job of responding to reviewer comments. I can see that they have gained a fuller understanding of the study limitations in particular, which they've addressed in their Discussion. I'm sure this will also help in the conduct and reporting of their research. This is an important study and I know others will be keen to learn more about it.